# Impact of the Parasitoids *Anisopteromalus calandrae* (Howard) and *Lariophagus distinguendus* (Förster) on Three Pests of Stored Rice

**DOI:** 10.3390/insects14040355

**Published:** 2023-04-03

**Authors:** Jordi Riudavets, Consuelo Belda, Cristina Castañé

**Affiliations:** IRTA, Sustainable Plant Protection, Ctra. Cabrils Km 2, 08348 Cabrils, Spain

**Keywords:** stored products, rice, biological control, fecundity, host suitability

## Abstract

**Simple Summary:**

Various pest species cause significant damage to stored rice, and polyphagous parasitoids could offer an appropriate biological control strategy. Among the beneficial species colonising silos and warehouses of rice, the generalist pteromalid parasitoids *Anisopteromalus calandrae* and *Lariophagus distinguendus* have been found to be prevalent in samples collected in the northeast of Spain. In this study, the Spanish strains of both parasitoids have shown that they may play important roles in controlling three coleopteran pests of stored rice: *Sitophilus oryzae, Rhyzopertha dominica* and *Lasioderma serricorne*. They reduced the number of pests that emerged due to effective parasitism (parasitoids reaching the adult stage and emerging from the grain), combined with other sources of host mortality, such as parasitoid immature mortality and parasitoid host feeding. Both parasitoids preferentially attack *S. oryzae*, which is a key pest of stored rice, but they also play an important role in the control of *R. dominica*, a less important pest. Although both parasitoids could consume *L. serricorne* larvae, their efficacy is low, and more specific natural enemies should be evaluated for the control of this pest.

**Abstract:**

This study evaluated the ability of pteromalid parasitoids *Anisopteromalus calandrae* and *Lariophagus distinguendus* reared on *Sitophilus zeamais* to control stored product coleopteran pests *Sitophilus oryzae*, *Rhyzopertha dominica* and *Lasioderma serricorne*. In trials of parasitoid treatment with *A. calandrae*, fewer pests (*S. oryzae* and *R. dominica*) emerged than in the control. Parasitoid reproduction was highest with *S. oryzae* as a host, followed by *R. dominica* and *L. serricorne*. In trials of parasitoid treatment with *L. distinguendus*, fewer pests (*S. oryzae*, *R. dominica* and *L. serricorne*) emerged than in the control treatment. *Sitophilus oryzae* was the host with the highest rate of parasitoid reproduction, although the greatest level of reduction was seen in *R. dominica* (i.e., host feeding levels were higher for this host species). For *L. serricorne*, no *L. distinguendus* progeny was produced. For both species, parasitoids with significantly longer bodies and tibiae emerged from *S. oryzae*. These results suggest that both parasitoids have potential for use as biocontrol agents for different coleopteran species that attack stored rice.

## 1. Introduction

Stored products are attacked by many species of pests. Today, they are controlled primarily using residual insecticides and fumigants. However, the development of pest resistance to many pesticides and the reduced availability of permitted active ingredients for chemical pest control has motivated the search for effective biological alternatives [1].

In Spain and other countries in the Mediterranean Basin, the most abundant and widely distributed primary pest species in stored rice are *Sitophilus zeamais* Motschulsky and *Sitophilus oryzae* (L.) (Coleoptera: Curculionidae) [2,3,4,5,6]. *Rhyzopertha dominica* (Fabricius) (Coleoptera: Bostrychidae) has also been identified as a serious pest of stored rice [7,8], the importance of which is increasing due to rising temperatures caused by climate change [9]. *Sitophilus* spp. appear earlier (spring and early summer) and are more abundant than *R. dominica*, which appear later in the year when temperatures are high. Other coleopteran pest species infest silos later, after the grain has been destroyed by primary pests. Among these secondary pests, *Lasioderma serricorne* (Fabricius) is commonly found in rice [10] and is one of the first species to colonise stores of previously damaged grain [11].

Due to the range of pest species that are simultaneously present in stored rice facilities, polyphagous natural enemies could offer an appropriate biological control option [12,13]. A generalist natural enemy can control different pest species to different degrees, depending not only on the characteristics of the natural enemy [14] but also on the type and abundance of host species present. The effectiveness of these enemies may also vary; certain strains could be better adapted to the local environment of a specific region. All these factors interact and modify the impact of natural enemies on pest control [15]. Among the beneficial species colonising non-treated silos and warehouses of rice, the generalist pteromalid parasitoids *Anisopteromalus calandrae* (Howard) and *Lariophagus distinguendus* (Förster) have been found to abound in samples collected in the northeast of Spain [3]. Both are solitary primary ectoparasitoids that attack late-instar larvae that develop within the grain kernel [16,17,18,19]. The spectrum of known hosts of these two parasitoids includes *S. oryzae* [20,21,22], *S. granarius* (L.) [23,24], *S. zeamais* [25], *R. dominica* [8,26,27,28] and *L. serricorne* [29,30], among others.

Timokhov and Gokhman [31] established that two previously considered strains of *A. calandrae* were in fact two sibling species. These species have different host preferences and life strategies; one is better adapted to parasitise Curculionidae and Bruchidae, and the other prefers to parasitise Anobiidae. Individuals from our *A. calandrae* laboratory colony were identified as the ‘real’ *A. calandrae* (Howard), which is characterised by a haploid chromosome number of 7 and by a strong preference for parasitising Curculionidae and/or Bruchidae [31]. There was no need to confirm the species of *L. distinguendus* as there was no uncertainty over its identification.

*Anisopteromalus calandrae* and *L. distinguendus* have been reared in our laboratory for more than four years, and their ability to control *S. zeamais* has been thoroughly evaluated [8,25]. The objective of the present study was to evaluate the ability of the Spanish strains of both parasitoids to control three main coleopteran pests of stored rice: *S. oryzae*, *R. dominica* and *L. serricorne*. The size of the offspring was also evaluated since larger parasitoids have higher parasitism rates and an increased ability to control pests [32].

## 2. Materials and Methods

### 2.1. Insect Colonies

Insect colonies were reared in the laboratory under controlled conditions (25 ± 2 °C; 70 ± 10% RH). The colonies were started with adults collected from several rice silos in the Barcelona region of northeastern Spain and periodically refreshed. Laboratory colonies of *S. oryzae*, *R. dominica* and *L. serricorne* were reared on standard media (*S. oryzae* and *R. dominica* on brown rice and *L. serricorne* on coarsely ground brown rice) [33]. The parasitoids *A. calandrae* and *L. distinguendus* were reared on *S. zeamais* larvae that developed in brown rice.

### 2.2. Control of Three Coleopteran Species by A. calandrae and L. distinguendus

To obtain separated populations containing all larval stages of each of the three hosts, the rice was infested with 600 adults/800 gr and kept in 2 L ventilated plastic containers under controlled conditions (25 ± 2 °C and 70 ± 10% RH). Three hundred *S. oryzae* or *R. dominica* adults were added fortnightly. For *L. serricorne*, 150 adults were added weekly because this species has a shorter lifespan than the other two hosts.

After five weeks, the adult beetles were sieved out, and 100 g of rice from each host population was placed in 0.5 L ventilated glass jars. Seven female and three male *A. calandrae* (less than 7 days old) were introduced to each jar. In the case of *L. distinguendus*, three females and one male (less than 7 days old) were introduced to each jar due to the low availability of individuals. Two Eppendorf tubes (Eppendorf Ibérica S.L.U., San Sebastián de los Reyes, Madrid, Spain) with sucrose solution and a cotton plug were added to all jars to feed the wasps. The parasitoids were kept in the jars for one week and then removed. During the following seven weeks, the numbers of host adults and parasitoids that emerged from the hosts were recorded weekly. The parasitoids were collected, sexed and stored in 70% alcohol. The length of the right hind tibia and the total body length from the top of the head to the end of the abdomen, excluding the ovipositor protrusion, of 20 females and 20 males from each species that emerged from each of the different hosts were measured using a stereomicroscope (Carl Zeiss Meditec Iberia S.A.U., Tres Cantos, Madrid, Spain) at 40× [34]. In the control treatments, the three hosts were placed in jars, but no parasitoids were released. Six replicate trials of each parasitoid and host combination were performed.

Host mortality measured as the difference between adult emergence in each parasitoid treatment compared to emergence in their respective control treatment, the number of parasitoid adult progeny (effective parasitism) and parasitoid size and sex ratio were used as response variables to assess host suitability [19]. The ratio of parasitoid-induced mortality (PIM) in the hosts (i.e., host mortality caused by the presence of the parasitoid that did not result in adult parasitoid emergence) [35] to the total number of hosts that emerged from the control treatment was also calculated.

### 2.3. Statistical Analyses

Before statistical analyses, the homogeneity of variances was tested using Bartlett’s test, and data sets were transformed when necessary. The number of *S. oryzae*, *R. dominica* and *L. serricorne* that emerged in the trials with each parasitoid species was compared to the number that emerged in the control treatments by a two-way ANOVA. The size (tibia and body length) of adult parasitoids of each sex that emerged from the different hosts was also compared by a two-way ANOVA. The number of *A. calandrae* and *L. distinguendus* adults that developed in each host species, the percentage of reduction in host emergence, the percentage of effective parasitism and the PIM caused by the two parasitoids were analysed using a one-way analysis of variance (ANOVA). Post hoc comparisons were conducted using the Tukey correction for multiple comparisons. All statistical analyses were conducted using JMP 16.2.0 (SAS Institute Inc., Cary, NC, USA, 2020–2021).

## 3. Results

### 3.1. Host Suitability Trials with A. calandrae

Although we infested brown rice with the same number of adults of each host, the number of progeny produced by each species was significantly different (*F* = 702.17; df = 5.35; *p* < 0.001): More *S. oryzae* adults emerged than *R. dominica* or *L. serricorne* adults (Figure 1A). In the presence of parasitoids, significantly fewer *S. oryzae* and *R. dominica* adults emerged compared to the control treatment without parasitoids (*S. oryzae*: *t* = 721.76, df = 1, *p* < 0.001; *R. dominica*: *t* = 93.07, df = 1, *p* < 0.001). However, the numbers of *L. serricorne* that emerged from the control and parasitoid treatments did not differ significantly (*t* = 0.645; df = 1; *p* = 0.4281).

The number of *A. calandrae* that emerged from each host species tested differed significantly (*F* = 61.10; df = 2, 17; *p* < 0.001). The largest number of descendants emerged from *S. oryzae*; fewer than half as many emerged from *R. dominica* as from *S. oryzae*. The number of parasitoid offspring collected from *L. serricorne* was very low (Figure 1B). The sex ratio of parasitoid offsprings was 52% females for *S. oryzae*, 45% females for *R. dominica* and 54% females for *L. serricorne.*

The parasitoid treatment reduced the number of pests that emerged due to effective parasitism (parasitoids reaching the adult stage and emerging from the grain) combined with other host mortality causes, such as parasitoid immature mortality and parasitoid host feeding (PIM) (Table 1). Higher percentages of total mortality were observed for *R. dominica* and *S. oryzae* (62% and 47%, respectively); *L. serricorne* had the lowest total mortality (10%). In *R. dominica* and *S. oryzae,* host mortality was due primarily to effective parasitism (57% and 36%, respectively), while in *L. serricorne*, mortality was due mainly to PIM (7.8%). There were significant differences in the percentage of total mortality and in the effective parasitism that the parasitoid produced in the three host species tested but not in the percentage of PIM.

Male and female parasitoids developed in *S. oryzae* and in *L. serricorne* larvae were larger than those developed in *R. dominica*; that is, they had longer tibiae and longer bodies. All females had longer tibiae and bodies than the males (Table 1).

### 3.2. Host Suitability Trials with L. distinguendus

Again, although we infested brown rice with the same number of adults of each host, the different species produced significantly different numbers of progeny (*F* = 167.39; df = 5.35; *p* < 0.001). More *S. oryzae* than *L. serricorne* adults emerged, and even fewer *R. dominica* adults emerged (Figure 2A). Nevertheless, in the treatment with *L. distinguendus*, significantly fewer adult *S. oryzae*, *R. dominica* or *L. serricorne* emerged than in the control treatment (*S. oryzae*: *t* = 46.97, df = 1, *p* < 0.001; *R. dominica*: *t* = 139.76, df = 1, *p* < 0.001; *L. serricorne*: *t* = 3.90, df = 1, *p* < 0.05).

This parasitoid was able to reproduce only in *S. oryzae* and *R. dominica*. No adult parasitoids emerged from the *L. serricorne* larvae (Figure 2B). Significantly different total numbers of *L. distinguendus* emerged from these two pest species (*F* = 228.54; df = 2.17; *p* < 0.001); *S. oryzae* produced the most adult parasitoids. The sex ratio in both host species was 64% females for *S. oryzae* and 53% females for *R. dominica*.

There were significant differences in the total percentage of host mortality and in the percentage of mortality that was due to effective parasitism or PIM in each host species. The highest total mortality was observed in *R. dominica* (73%), which experienced approximately twice as high mortality as *L. serricorne* or *S. oryzae* (37% and 30%, respectively). Nearly half of the mortality in *R. dominica* was due to effective parasitism and half to PIM; similarly, in *S. oryzae*, approximately half of the total mortality was due to parasitism and half to PIM. However, in *L. serricorne,* mortality was due entirely to PIM since no adult parasitoids emerged (Table 2).

Male and female parasitoids developed in *S. oryzae* larvae were larger than those that developed in *R. dominica*; also, in both host species, female parasitoids had longer tibiae and body lengths than males (Table 2).

## 4. Discussion

In the experiment conducted with each parasitoid, the host *S. oryzae* produced many more progenies in the control treatment than did *R. dominica* (3.8 times as many with *A. calandrae* or 19 times as many with *L. distinguendus*) or *L. serricorne* (4.6 times as many with *A. calandrae* or 4.7 times as many with *L. distinguendus*) (Figure 1A and Figure 2A). This occurred even though the rice was infested with a similar number of adults for each host tested, and we used the same amount of infested rice for each host. Therefore, *S. oryzae* reproduced most successfully under the conditions of our experiments: with brown rice and at 25 ± 2 °C and 70 ± 10% RH. The opposite results were observed when *S. zeamais* and *R. dominica* were reared in paddy rice under the same environmental conditions as those in the present experiments: *R. dominica* reproduced twice as much in paddy rice as did *S. zeamais* (1.45 vs. 0.69 adults/gr of rice, respectively) [36]. Since *R. dominica* is a grain borer, it can likely penetrate the rough skin of the paddy rice better than the smooth surface of the brown rice; the opposite is likely true of the *Sitophilus* species tested. These results indicate that problems with the *Sitophilus* species could be accentuated when rice is stored in poor conditions and contains many unhusked grains or after the rice is polished and packaged rather than stored as paddy rice in good storage conditions. Both parasitoids reproduced more than twice as much in *S. oryzae* as in *R. dominica* or *L. serricorne*. This reflected the abundance of hosts offered to the females for oviposition, as demonstrated by the number of adults produced in the control treatment for each host species (Figure 1B and Figure 2B).

Although very different numbers of *S. oryzae* and *R. dominica* were offered to *A. calandrae* (2200 vs. 650, respectively), the percentage of emerged hosts was not significantly lower than that in the control treatment (Table 1). This indicates that the parasitoid controlled *S. oryzae* more effectively than *R. dominica*. If we had offered a similar number of both hosts, we would likely have observed a larger reduction in *S. oryzae* than in *R. dominica.* The adult parasitoid progeny was also larger when they developed in *S. oryzae* larvae than in *R. dominica* larvae (Table 1). This also indicates that the former is a more suitable host than the latter, since larger females usually produce more eggs. This is likely related to the size of the larvae offered, as *R. dominica* larvae are smaller than *Sitophilus* larvae. Ghimire and Phillips [19] also observed that *A. calandrae* females and males were significantly larger and heavier when they developed in *S. oryzae* than in *R. dominica*. A preference for *S. zeamais* was observed when different proportions of this species and *R. dominica* larvae were offered to *A. calandrae* females: in all proportions of the two host species offered, reproduction was significantly higher in *S. zeamais* than in *R. dominica* [36], indicating that the parasitoid has a clear preference for *S. zeamais*.

*Anisopteromalus calandrae* females used both *S. oryzae* and *R. dominica* larvae primarily for reproduction. The effective parasitism in both species accounted for more than 75% of the total host mortality, and only a small proportion of host mortality was due to PIM. In the present study, *A. calandrae* females did not successfully reproduce in *L. serricorne* larvae, since only a few adult parasitoids were produced. In contrast, Guo et al. [37] report that *A. calandrae* has great potential for controlling *L. serricorne* infestations. This difference is likely because the individuals of *A. calandrae* used in our study belong to the strain that is known to prefer to parasitise Curculionidae and/or Bruchidae [31] and are less successful at parasitising the larvae of Coleoptera from other families, such as Anobiidae (*L. serricorne*).

*Lariophagus distinguendus* followed a similar pattern in our study, even though this wasp was released at a lower ratio than *A. calandrae*. Although this parasitoid caused a significantly higher percentage of mortality in *R. dominica* than in *S. oryzae* (Table 2), significantly different numbers of hosts were offered (127 *R. dominica* vs. 1893 *S. oryzae*). Again, this parasitoid’s effectiveness in controlling the latter is important due to the large numbers of larvae attacked and parasitoids offspring produced (Figure 2B). The parasitoid progeny was also larger when they developed in *S. oryzae* than in *R. dominica* (Table 2), indicating that the former species is a better host for parasitoid reproduction than *R. dominica*. In this case, effective parasitism accounted for approximately the same percentage of mortality as PIM in both hosts (Table 2). However, although the parasitoid caused 37% mortality in *L. serricorne* larvae, no adult parasitoids emerged, indicating that this is not an appropriate host for parasitoid reproduction. In contrast, Jiménez-Ambriz et al. [30] showed that extracts of *L. serricorne* cocoons were attractive to this parasitoid, and Papadopoulou and Athanassiou [38] described the successful development of this parasitoid in *L. serricorne*. Differences in fecundity [24] or the long period of laboratory rearing in our study could account for these differences in our findings, as similar discrepancies have been observed with other pteromalid species [39]. In fact, our laboratory rearing process was not very successful, as indicated in the methodology.

## 5. Conclusions

Both parasitoids preferentially attack *Sitophilus* spp., which are key pests of stored rice, but they also play an important role in the control of *R. dominica*, a less important pest. Although both parasitoids can parasitise *L. serricorne* larvae, their efficacy is low, and more specific natural enemies should be evaluated for their control. While *L. distinguendus* has been found during several screenings of storage grain in Spain, it has always been much less abundant than *A. calandrae* [11,40]. This could be because *A. calandrae* is better adapted to hotter conditions than *L. distinguendus,* which is preferentially found in central and northern Europe [41]. Therefore, the Spanish strains of both parasitoids could play important roles in controlling the primary pests of stored rice.

## Figures and Tables

**Figure 1 insects-14-00355-f001:**
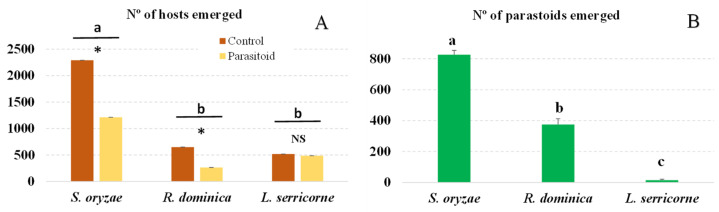
Number (mean + SEM) of emerged host progeny (**A**) and of *A. calandrae* adults that developed in each host species (**B**). Bars for each host headed by a different letter and bars within each host headed by an asterisk are significantly different (*p* < 0.05, Tukey test), NS indecates no significant difference.

**Figure 2 insects-14-00355-f002:**
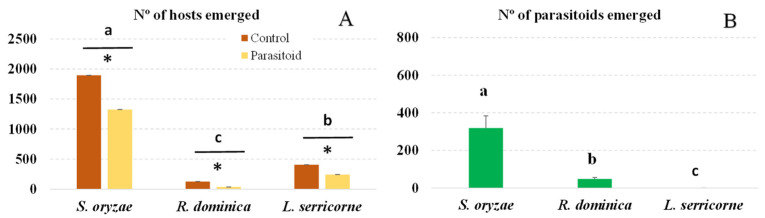
Number (mean + SEM) of emerged host progeny (**A**) and of *L. distinguendus* adults that developed in each host species (**B**). Bars for each host headed by a different letter and bars within each host headed by an asterisk are significantly different (*p* < 0.05, Tukey test), NS indecates no significant difference.

**Table 1 insects-14-00355-t001:** Percentage (mean ± SEM) of host mortality, parasitism and PIM (host feeding, unsuccessful parasitism) caused by *A. calandrae* when the larvae of *S. oryzae*, *R. dominica* or *L. serricorne* were offered. Mean (±SEM) hind tibia and body length of *A. calandrae* (males and females) emerging from each host species. N = 40 (20 males and 20 females). In each row, means followed by a different letter are significantly different (*p* < 0.05, Tukey). Body size measurements are given in mm.

Variables	*S. oryzae*	*R. dominica*	*L. serricorne*	*F*	*df*	*p*
Total mortality (%)	47.0 ± 1.5 a	62.3 ± 5.1 a	10.4 ± 3.5 b	50.58	2.17	<0.001
Parasitism (%)	36.1 ± 1.2 b	57.7 ± 5.7 a	2.8 ± 1.2 c	78.68	2.17	<0.001
PIM (%)	10.6 ± 1.3 a	4.6 ± 3.9 a	7.8 ± 3.7 a	2.30	2.17	0.13
Hind tibia length 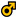	0.42 ± 0.003 a	0.34 ± 0.013 b	0.40 ± 0.013 a	47.12	5.119	<0.001
Hind tibia length 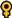	0.52 ± 0.006 a	0.41 ± 0.011 b	0.50 ± 0.010 a
Body length 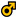	1.63 ± 0.020 a	1.21 ± 0.035 b	1.73 ± 0.044 a	79.81	5.119	<0.001
Body length 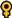	2.12 ± 0.034 a	1.58 ± 0.032 b	2.08 ± 0.053 a

**Table 2 insects-14-00355-t002:** Percentage (mean ± SEM) of host mortality, effective parasitism and PIM (host feeding, unsuccessful parasitism) caused by *L. distinguendus* when the larvae of *S. oryzae*, *R. dominica* or *L. serricorne* were offered. Mean (±SEM) hind tibia and body length of *L. distinguendus* (males and females) emerging from each host species. N = 40 (20 males and 20 females). In each row, means followed by a different letter are significantly different (*p* < 0.05, Tukey). Body size measurements are given in mm.

Variables	*S. oryzae*	*R. dominica*	*L. serricorne*	*F*	*df*	*p*
Total mortality (%)	30.0 ± 6.3 b	73.6 ± 9.8 a	37.1 ± 10.4 b	6.26	2.17	<0.01
Parasitism (%)	17.0± 3.4 a	35.5 ± 7.3 a	0 b	20.87	2.17	<0.001
PIM (%)	13.0 ± 3.2 b	38.1 ± 6.3 a	37.1 ± 10.4 a	4.07	2.17	0.04
Hind tibia length 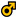	0.40 ± 0.01 a	0.29 ± 0.01 b	-	36.65	3.79	<0.001
Hind tibia length 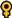	0.47 ± 0.01 a	0.32 ± 0.01 b	-
Body length 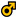	1.33 ± 0.04 a	1.0 ± 0.04 b	-	52.43	3.79	<0.001
Body length 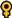	1.81 ± 0.04 a	1.2 ± 0.05 b	-

## Data Availability

The data presented in this study are available upon request from the corresponding author.

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
