# Peer review of "Impact of the Parasitoids Anisopteromalus calandrae (Howard) and Lariophagus distinguendus (Förster) on Three Pests of Stored Rice"

_insects, 2023, doi:10.3390/insects14040355_

Round 1

Reviewer 1 Report

The submitted paper has high degree of novelty, provides interesting and applicable results, which presented in clear, concise, coherent and replicable manner. Introduction is closely related to the subject, Material and methods describes clearly and are replicable, results very well interpreted and Discussion related to the results. the literature cited is adequate.

Therefore, I recommend this paper for publication in this journal after correction of several grammatical changes and two dileammas in Discussion section, that are submitted in a separate file.      

Author Response

No specific questions are posed by the reviewer to the authors, just his/her recommendation to the editor.

Reviewer 2 Report

The manuscript represents the use of polyphagous parasitoids A. calandrae and L. distinguendus  as a biological strategy for controlling serious pests of stored rice. The topic is relevant and results contribute to newer knowledge of the application of natural enemies as control strategy for stored product pests. The manuscript is well-structed, with appropriate experimental design. Data are clear and interpreted appropriately. There are some minor suggestions for the changes to the results display. After the authors make changes according to the suggested recommendation (listed within the attached file), the manuscript would be suitable for the publications in the journal Insects.

COMMENTS AND SUGGESTIONS

Line 41 Write the full name of the species at the first mention in the section: Sitophilus zeamais and Sitophilus oryzae

Line 128 Word More write with the small initial letter

Line 146 Put the full stop after L

Line 159 The first sentence in the line is redundant here, it is already explained in the section Materials and Method

Line 162/163 The last sentence does not belong to the explanation title of the table. It would be more suitable to put it within the section Results

Table 1. The unit of measurement for tibia and body length should be specified (also in the Table 2)

Line 168 Put the full stop after S

Line 195 The same comment as for the Line 159

Line 198/199 The same comment as for the Lines 162/163

Conclusion: I suggest that the conclusion end with authors own statement rather than another reference

Reference: Since only 15% of the refences used were published in the last 5 years, authors are recommended to use more recent references

Author Response

COMMENTS AND SUGGESTIONS

Line 41 Write the full name of the species at the first mention in the section: Sitophilus zeamais and Sitophilus oryzae. DONE

Line 128 Word More write with the small initial letter DONE

Line 146 Put the full stop after L DONE

Line 159 The first sentence in the line is redundant here, it is already explained in the section Materials and Method. DONE

Line 162/163 The last sentence does not belong to the explanation title of the table. It would be more suitable to put it within the section Results. The reviewer is right, and the information is already included in the Results Section (Line 164). So, we deleted the sentence in the caption of the table.

Table 1. The unit of measurement for tibia and body length should be specified (also in the Table 2). DONE

Line 168 Put the full stop after S. DONE

Line 195 The same comment as for the Line 159. DONE.

Line 198/199 The same comment as for the Lines 162/163. The reviewer is right, and here again the information is already exposed in the Results Section (Lines 200-201). So, we deleted the sentence in the caption of the table.

Conclusion: I suggest that the conclusion end with author’s own statement rather than another reference. DONE.

Reference: Since only 15% of the refences used were published in the last 5 years, authors are recommended to use more recent references. We have replaced four references related with general aspects of the topic with four more up-to-date articles.

Reviewer 3 Report

Riudavets et al. present the result of a simple laboratory experiment designed to examine the reproduction of two pteromalids on three coleopteran pests of rice in Spain. The results are clear and will be of interest to readers of Insects.

I believe the manuscript can be improved. I have written numbered points and suggestions for improvements to the text on a scanned copy of the manuscript.

Numbered points (see scanned manuscript)

1. "Chemicals" is a vague term but implies synthetic insecticides. Is this correct? I would have thought that biorrational products (e.g. Bt, spinosad, etc.) were used in grain silos. Or are you talking of fumigants? Please be more precise.

2. Reword. ...of permitted active ingredients for chemical pest control has motivated the search for effective biological alternatives [1].

3. Please give full species names at first use.

4. A pers. observation by whom?

5. Strategy is not the term here.  Change to "option" or "tool". (Strategy would be inundative/inoculative biocontrol).

6. Please describe the laboratory conditions under which the experiment was performed (not mentioned until the Discussion).

7. Please make it clear that each host was reared separately and each parasitoid was tested separately.

8a. Age of parasitoids used?

8b. The way it is written, it sounds like you mixed two parasitoid species in each jar.  Reword.

8c. Recorded how often? Daily?

8d. Describe the points on the body used to define body length measurements.

9. It was unclear to me whether the parasitoids emerged and were collected, or whether they emerged and engaged in parasitism and were then collected. Two generational cycles are mentioned in the text. Please clarify if possible.

10. How did you measure host mortality? I could not find info on counting dead/parasitised hosts. Was this inferred from the numbers of control hosts?

11. Reword: i.e. host mortality caused by the presence of the parasitoid that did not result in adult parasitoid emergence.

12. You present sex ratio as a proportion (proportion female???) rather than a F:M ratio.

Table 1. Table 2. Are body measurements given in mm?

13. Did these parasitism data meet the Bartlett's test for homoscedasticity?

14. Were two experiments performed? Or one experiment with two species?

Author Response

Riudavets et al. present the result of a simple laboratory experiment designed to examine the reproduction of two pteromalids on three coleopteran pests of rice in Spain. The results are clear and will be of interest to readers of Insects.

I believe the manuscript can be improved. I have written numbered points and suggestions for improvements to the text on a scanned copy of the manuscript. ANSWER: We thank all his/her suggestions that we have followed.

Numbered points (see scanned manuscript)

  1. "Chemicals" is a vague term but implies synthetic insecticides. Is this correct? I would have thought that biorrational products (e.g. Bt, spinosad, etc.) were used in grain silos. Or are you talking of fumigants? Please be more precise. ANSWER: We have replaced the term chemicals by "insecticides and residual fumigants". Line 37.
  2. Reword. ...of permitted active ingredients for chemical pest control has motivated the search for effective biological alternatives [1]. ANSWER: Modified as suggested. Lines 38-39.
  3. Please give full species names at first use. DONE. Lines 41-42.
  4. A pers. observation by whom? ANSWER: Authors personal observation. Modified and indicated according to the journal's Instructions for Authors guide. Line 49.
  5. Strategy is not the term here.  Change to "option" or "tool". (Strategy would be inundative/inoculative biocontrol). ANSWER: Changed to “option”. Line 52.
  6. Please describe the laboratory conditions under which the experiment was performed (not mentioned until the Discussion). ANSWER: Added. They were the same conditions as for the insect rearing. Line 82.
  7. Please make it clear that each host was reared separately and each parasitoid was tested separately. ANSWER: We added “separately”. Line 91.

8a. Age of parasitoids used? ANSWER: Less than one week old. Added.

8b. The way it is written, it sounds like you mixed two parasitoid species in each jar.  Reword. ANSWER: We have reworded the sentence in lines 97-100, and now we believe it is clearer.

8c. Recorded how often? Daily? ANSWER: Weekly. Added. Line 103.

8d. Describe the points on the body used to define body length measurements. ANSWER: We measured the total body length from the top of the head to the end of the abdomen, excluding the ovipositor protrusion. Added.

  1. It was unclear to me whether the parasitoids emerged and were collected, or whether they emerged and engaged in parasitism and were then collected. Two generational cycles are mentioned in the text. Please clarify if possible. ANSWER: Two sentences earlier, we explained that the parasitoids were collected every week, so they were not left in the jars. We agree with the reviewer that these two sentences may be confusing to the reader and have removed them.
  2. How did you measure host mortality? I could not find info on counting dead/parasitised hosts. Was this inferred from the numbers of control hosts? ANSWER: We measured host mortality as the difference between adult host emergence in each parasitoid treatment compared to the emergence in the control treatments. Added. Line 105.
  3. Reword: i.e. host mortality caused by the presence of the parasitoid that did not result in adult parasitoid emergence. DONE. Line 114-115.
  4. You present sex ratio as a proportion (proportion female???) rather than a F:M ratio. ANSWER: Both sentences were amended accordingly. Line 143-145. Lines 183-184.

    Table 1. Table 2. Are body measurements given in mm? ANSWER: Yes. This is now indicated in the title of the table.

    1. Did these parasitism data meet the Bartlett's test for homoscedasticity? ANSWER: As explained in the section 2.3. “Statistical analyses”, the homogeneity of variances was tested using the Bartlett’s test before performing the statistical analyses. Now, this sentence has been moved to the start of the paragraph (lines 119-120).
    2. Were two experiments performed? Or one experiment with two species? ANSWER: We conducted one experiment with each parasitoid species. Line 210.

Round 2

Reviewer 3 Report

The authors have addressed my concerns.